# Delusion progression process from the perspective of patients with psychoses: A descriptive study based on the primary delusion concept of Karl Jaspers

Naoki Hayashi[1]*, Yoshito Igarashi[2]°, Hirohiko Harima[3]°

1 Department of Psychiatry, Teikyo University School of Medicine, Tokyo, Japan, 2 Division of Law and Psychiatry, Center for Forensic Mental Health, Chiba University, Chiba, Japan, 3 Department of Psychiatry, Tokyo Metropolitan Matsuzawa Hospital, Tokyo, Japan

° These authors contributed equally to this work.

* nhayashi55@nifty.com

## Abstract

**Data Availability Statement:** All relevant data are within the paper and its Supporting Information files.

### Background

Delusion occupies an important position in the diagnosis and treatment of patients with psychoses. Although Karl Jaspers' concept of the primary delusion (PD) is a key hypothesis in descriptive phenomenology concerning the primordial experience of delusion, to our knowledge it has not been verified in empirical studies of patients with psychosis, and the relationship between PDs and fully developed delusions remains unclear.

### Methods

The subjects were 108 psychiatric patients diagnosed with DSM-IV schizophrenia or schizoaffective disorder who had persisting delusions. This investigation used a newly devised semi-structured interview, the Delusion and its Origin Assessment Interview (DOAI), and the Positive and Negative Syndrome Scale. PDs enquired about in the DOAI were delusional perception, delusional memory, delusional mood, and delusional intuition. Associations of PDs with delusion themes and delusion features extracted from DOAI items by factor analysis were examined using correlational and MANCOVA regression analyses. Reliability studies of the DOAI were also conducted.

### Results

The reliability and correlation analyses suggested robust psychometric properties of the DOAI. The percentages of subjects reporting PD phenomena as delusion origins and currently present were 93% and 84%, respectively. MANCOVA revealed several significant associations, including between delusional perception and delusional mood and persecutory themes, between delusional intuition and grandiose delusions, and between delusional perception and intuition and systematization of delusions.

**Funding:** This study was supported by a Grant-in-aid for exploratory research (no. 11877164) from the Japan Society for the Promotion of Science. The funders had no role in study design, data collection and analysis, decision to publish, or preparation of the manuscript.

**Competing interests:** The authors have declared that no competing interests exist.

## Discussion

This study demonstrates that PDs can be considered as principal origins of delusions by subjects with psychosis, and have meaningful connections with the characteristics of their fully developed delusions. The associations between PDs and delusion characteristics can be interpreted in terms of progression processes of delusions, which are seen as intensification and generalization of cognitive and affective pathologies in PDs. The findings are also consistent with the neurobiological hypothesis that aberrant salience attribution to stimuli, as in PDs, is the primary phenomenon caused by abnormal dopamine system regulation. Further studies are needed to clarify delusion progression processes relating to PDs and to substantiate their clinical meanings.

## Introduction

Delusion constitutes a core aspect of psychiatric symptomatology and is essential in diagnostic practice for patients with psychotic disorders [1–7]. Processes of delusion formation and dissolution are also of considerable interest in clinical research and treatment [2, 3, 8]. However, capturing the psychological changes that form a background to delusions is often considered infeasible [9–11], usually because delusions are thought to be influenced by diverse disturbances of mental activities such as thinking, perception, memory, and affect [2, 4, 6, 7, 12, 13].

Descriptive psychopathology has historically provided the framework for the investigation of delusions [1, 2, 5, 7, 11]. Among others, Jaspers laid down the methodology of the phenomenological approach for describing in-depth patients' subjective experiences (phenomena) [3, 5, 11]. He defined methods for understanding (comprehending; Verstehen) such experiences. The first was static understanding, i.e., to vividly recreate a patient's experiences within the observer's mind while thoroughly excluding presuppositions. The next was genetic understanding, which was to seek out psychic connections with other experiences or events the patients presented by empathizing or putting oneself in the patient's situation. He also noted the limits to his method of understanding pathological experiences, and introduced the distinction between understandability (comprehensibility; Verständlichkeit) and un-understandability of the experience from the viewpoint of the observer [12, 14]. Based principally on this methodology, Jaspers defined delusions by listing their basic features: extraordinary conviction, incorrigibility, and impossibility of content, and gave a comprehensive description of delusions and other psychotic experiences. He added that un-understandability of the experiences was characteristic of patients with schizophrenia.

By utilizing the accumulated efforts of Jaspers and other authors, Schneider advanced the description of psychotic experiences and wove delusional symptoms into his psychiatric nosology [3, 11, 15]. His hallmark achievement was to delineate first rank symptoms (FRSs), a list of particularly un-understandable psychotic symptoms, that he considered to be pathognomonic of schizophrenia [1]. Some FRSs would today be considered examples of bizarre delusions, as defined by the criteria for schizophrenia in the Diagnostic and Statistical Manual of Mental Disorders, 4th Edition (DSM-IV) [16]. The Schneiderian approach has also been partly incorporated into psychiatric evaluation systems such as the Present State Examination (PSE) [17], Das AMDP-System (AMDP) [18], and recent versions of the Examination of Anomalous Self-Experience (EASE) [19] and Examination of Anomalous World Experience (EAWE) [20].

Among current descriptive studies of psychotic experiences, Kendler et al. [21] and others [22–26] conducted a series of investigations to identify basic features of delusions usable for

comprehending the experience. Characteristics assessed in these studies included conviction, preoccupation, extension (generalization), affective response (anxiety, unhappiness, etc.) and systematization. In addition, Kendler et al. [21] and Oulis et al. [26] performed factor analytic studies on delusion characteristics, and extracted a number of principal features, such as those representing influence by delusion and disturbed thought process; however, the results of these two studies did not converge well. With the aim of bringing order to the vast variety of delusional symptoms, Spitzer [9] proposed a distinction between delusions about external reality and those concerning happenings within the patient's mind. One further framework for comprehending delusion is the ipseity model of Sass et al. [27], which is an attempt to grasp diverse psychotic symptoms by postulating alteration of the self-state as the basic underlying pathology.

Descriptive psychopathology has also attempted to elucidate developmental processes in delusions [11]. Jaspers postulated primary delusions (PDs) (primären Wahnerlebnisse) as a particular type of phenomenon, further phenomenological tracing back of which to their origins (genetic understanding) was impossible [12]. The PDs referred to by Jaspers [12] were delusional awareness or mood (atmosphere), delusional perception, and delusional idea (delusional intuition or sudden notion) and recollection of a delusion-confirming memory (Sims [6] later paraphrased this last PD as delusional memory, a term that Jaspers did not explicitly use). Jaspers maintained that PDs were typical of un-understandable phenomena, being immediately experienced by the patients as absolutely self-evident like an intuition or a revelation and accompanied by abnormalities in the form such as radical transformation or emergence of thinking. He also designated PDs as delusion proper (echte Wahnidee), from which secondary delusions evolve via conscious thought processes [12, 14, 28].

Of further relevance to Jaspers' concept of PDs is the detailed description of the onset stage of schizophrenia carried out by Conrad [29]. He postulated three early stages, of domination of the individual's thinking by ominous and diffuse anxiety (trema), followed by precipitation of near-psychotic experiences, and then the manifestation of overt psychotic symptoms. Based on this, Huber & Gross [3] positioned delusional mood in the first stage, and delusional perception in the second and third stages of Conrad. Markova & Berrios [30] outlined a formative process of psychotic symptoms from experiencing something unnamed to expressing them as the symptoms in verbal communication. Taking a cognitive psychology perspective, Garety et al. [13] constructed a model of delusion formation based on the assessment of the characteristics of delusions, in which abnormal cognition, emotion, and thought disorder promote their formation. Moritz et al. [31] also presented a two-stage cognitive model composed of liberal acceptance (belief formation) and consolidation by factors such as emotional state (belief perpetuation).

From these and other considerations, there emerged the hypothesis that the primordial conditions of delusion are linked to a particular neurobiological abnormality. By focusing on phenomenological features of delusional mood, Kapur [32] proposed the aberrant salience hypothesis that chaotic disinhibition of subcortical and prefrontal dopamine release in acute schizophrenic psychosis is associated with excessive salience being attributed to otherwise irrelevant stimuli. Subsequently, van Os [33] widened the scope of this theory, maintaining that aberrant salience attribution underlies a wide variety of psychotic symptoms.

Although the primordial conditions of delusion may be a promising area of investigation in descriptive psychopathology, the number of studies in this area is limited. To our knowledge, the PD concept of Jaspers has been excluded from the scope of empirical validation, and although it was referred to by a few authors [3, 6, 10], most of these studies treated each PD individually. Therefore, this classical concept deserves reappraisal from a present day perspective. Based on the method of describing patients' subjective experiences and self-reflections,

albeit only a restricted part of the methodology of Jaspers, the present study attempted to confirm whether PDs represent the initial phenomena of delusional experiences. Furthermore, to explore what connections individual PDs have with delusional themes and features, this study investigated the characteristics of delusional experiences. By pursuing answers to these questions, the study aimed to clarify the clinical relevance of PDs, and provide new insights into the formation and perpetuation of delusions.

## Methods

### Subjects

Patients receiving treatment at Tokyo Metropolitan Matsuzawa Hospital (TMMH), a large regional psychiatric center for psychiatric emergencies and other services in central Tokyo, were enrolled based on a list of potential candidates provided by psychiatrists working there. The criteria for being a candidate were (i) having a persisting delusion of a mild or more severe level, as rated on the delusion Item P1 of Positive and Negative Syndrome Scale (PANSS) [34], (ii) being judged as sufficiently stable to undergo the assessment and not having marked difficulties in ordinary interpersonal communication and personal care, (iii) not having a comorbid diagnosis of dementia and other cognitive disorders, or mental retardation, and (iv) providing consent to participate in this study.

Investigators confirmed that subjects fulfilled these criteria and obtained written informed consent for participation. Diagnoses were made according to DSM-IV criteria by examining case records [35] and additional diagnostic enquiries where necessary.

### Assessment of PDs and delusion characteristics

The Delusion and its Origin Assessment Interview (DOAI), is a semi-structured interview schedule that addresses characteristics of delusional experiences and PDs, plus experiences that have been alleged to be origins of delusions (or reasons for having delusions) in the three months prior to the assessment. The DOAI is composed of three sections: A. Delusional themes, B. PDs, and C. Response and judgments. This schedule is included in the supporting information of this study as S1 Text.

**Section A. Delusional themes.** In this section, the symptomatic severity of delusions over the past three months is assessed, covering individual themes. This section includes six items: A-1. Persecutory delusion, A-2. Grandiose delusion, A-3. Hypochondriacal delusion, A-4. Being assisted or loved (erotomanic) delusion, A-5. Delusion of guilt, and A-6. Delusion of jealousy. These themes were selected from the most frequent delusion themes in the study of Huber and Gross [3]. The items are rated on a 5-point scale of severity with anchor points (1. Absent or minimal, 2. Mild, 3. Moderate, 4. Moderately severe, and 5. Severe). Item scores for the four most frequent delusional themes were used in later analyses.

**Section B. PDs.** This section enquires about the four PD types that Sims [6] defined based on descriptions by Jaspers [12]: B-1. Delusional perception (D-Percep) (Wahnwahrnehmung), B-2. Delusional memory (memory recall) (D-Memo) (Wahnerinnerung), B-3. Delusional mood (D-Mood) (Wahnstimmung), and B-4. Delusional intuition (D-Intuit) (Wahneinfall). In the first part of this section, subjects are asked about the origins of their delusions or reasons for having them, and then are asked about the degree of explanation for each delusion by individual PDs as origins and their frequency in the few weeks before the study assessment. These items are rated on a five-point scale with anchor points (1. None, 2. Unclear, 3. Mild, 4. Moderate, and 5. High for the degree of explanation; and 1. None or very rare, 2. Approximately once a day (Rare), 3. Sometimes, 4. Often, and 5. Almost always for frequency).

D-Percep, assessed in Item B-1, is the experience that a true perception has a special or unusual meaning for the subject. In the assessment of this item, delusions of reference induced by actual perceptions were included as D-Percep. It should be noted that there is inconsistency among authors concerning the definition of D-Percep: Schneider [1] and his student Koehler [15] stressed that total unrelatedness between perception and its connected thoughts should be a prerequisite for D-Percep. Schneider [1] additionally regarded D-Percep as a symptom of extreme anomaly and included it as a FRS. In contrast, Jaspers [12] noted that some delusions of reference could be included as D-Percep if they were initiated by actual percepts. In the present study, the assessment was conducted according to Jaspers' broader definition because there were many cases in which it was difficult to determine the extent of unrelatedness [5, 29].

Item B-2. D-Memo assesses memory of events or experiences that occurred longer than three months before the assessment. If the remembered event or experience occurred within the three-month period, it was rated using other PD items. The reason for making this operational distinction was that it was considered difficult to differentiate D-Memo from memories of past D-Intui and D-Percep, or delusional interpretation of past events [1, 6]. In addition, although it has been contended that D-Memo can be classified into types in terms of whether the original memory is based on a real experience, i.e., delusional interpretation of memory or retrospective memory falsification, this distinction was not taken into account in the assessment due to the difficulty in differentiating the two types [1, 3, 6].

In Item B-3, D-mood, the experience of an ominous and diffuse feeling of something impending, ambiguous and not yet defined, is enquired about. Experiences referred to as "world destruction phantasy or delusion of catastrophe (Weltuntergangserlebnis)" were also included in this assessment, following Jaspers [12], unless they were connected with actual percepts or events.

Item B-4 enquiring about D-Intui, an experience of an intuition or suddenly occurring notion related to delusions, is rated when it is not explicable as being caused or initiated by perception, memory recall, mood, or other abnormal experiences such as hallucinations or self-disturbance.

The order of enquiring about the 4 types of PDs was determined by the feasibility of specifying the experiences for the subjects during assessment. Hallucinations were not dealt with as origins of delusions because Jaspers did not count them as primary experiences of delusion, although he accepted that they could form the basis of delusion-like ideas (secondary delusions) (wahnhafte Ideen) [12, 14].

**Section C: Response and judgment.** This section addresses the subjects' responses to and judgments about their delusions. The items are C-1. Affective responses (C-1-a. Unpleasantness, C-1-b. Anxiety and tension, and C-1-c. Excitement and anger), C-2. Preoccupation, C-3. Inability to stop delusional thoughts, C-4. Distress, C-5. Pervasiveness, C-6. Delusional system formation, and C-7. Lack of insight into delusions (C-7-a. Conviction, C-7-b. Shareability of delusional thoughts, C-7-c. Reality attribution, and C-7-d. Denying mental illness involvement).

Section C. Response and judgment items are rated on a 5-point scale with anchor points (1. Opposite, 2. Neutral, 3. Mild, 4. Moderate, and 5. Severe for Affective response items, and 1. None, 2. Slight, 3. Mild, 4. Moderate, and 5. Severe for the other items).

Item C1 assesses the level of affect aroused by delusional experiences. Negative affect related to delusions has been enquired about in previous studies: concern [21], worry and unhappiness [22, 23], anxiety [23], anger [23], and emotional impact [26]. In Item C2, the level of preoccupation with the experience is assessed. This characteristic has also been included in previous studies [21–26]. In Item C3, degree of inability to stop or be distracted from

delusional thoughts is assessed. Equivalent items have been rated in previous studies [22, 26]. In Item C4, the level of distress caused by the delusional experience is assessed. This characteristic is also enquired about in existing studies [22, 25]. In Item C5, the level of pervasiveness, i.e., the breadth of delusional persons or objects, is assessed. This characteristic has again been investigated in previous studies [22, 26]. In Item C6, the level of delusional elaboration and systematization is assessed, as in two previous studies [21, 26]. In Items C7-a to C7-d, the degree of conviction concerning the delusion, the assumption of being able to share delusional thoughts with others, and the attribution of the thoughts to reality or to the subject's imagination are enquired about. These items were considered to be related to lack of insight into delusions, i.e., ability to recognize delusions as symptoms of illness, formed by an internal sense of abnormality and relevant communication with the external environment [36]. Previous studies have also rated conviction [21–24, 26], shareability [26], and insight into delusions [23].

**Reliability studies of the DOAI.** To examine the test-retest reliability of the DOAI, a sub-sample of 22 subjects was interviewed twice, 7 to 10 days apart. An inter-rater reliability study was also conducted in which two raters (NH & YI) made independent ratings of DOAI items in a single interview of the 22 subjects.

## Assessment of psychotic symptoms

To assess psychotic symptoms, the PANSS was employed [34], which has been widely used for the assessment of patients with psychoses in clinical and research settings. Our group has previously reported good reliability of the Japanese version of the PANSS [37].

## Statistical analysis

First, the test-retest and inter-rater reliability of DOAI items were examined by calculating analysis of variance intraclass correlation coefficients (ANOVA ICCs).

Next, to construct the delusion feature scales, an explorative factor analysis based on maximum likelihood (ML)-extraction with Varimax rotation of Section C items was performed. The number of factors was decided based on the criterion of their having an eigenvalue greater than 1. A simple factor structure was achieved by eliminating items that did not have a factor loading greater than 0.5 with any factor. Subsequently, composite scales were constructed for items that pertained to each of the factors (i.e., items with a factor loading greater than 0.5). These composite scale scores were used for later analyses as delusion feature variables.

To examine the convergent validity and clinical significance of DOAI scales of PDs and delusional themes and features, correlational analyses of these scores with those of the consensus PANSS factors proposed by Wallwork et al. [38], and also with relevant PANSS individual items. Pearson's product-moment or Spearman's rank-order correlation coefficients were used as appropriate.

Lastly, to identify associations between PDs and delusion themes and features, regression analyses applying multivariate analysis of covariance (MANCOVA) to delusion themes and features with the presence of the four types of PDs as factors, and sex and age (years) as covariates, were conducted. In these analyses, PD variables (B-1 to B-4) were collapsed into two values: lower than 3 (Mild) = 0, and equal to or higher than 3 = 1.

The software packages of SPSS 24.0.0.0 (IBM, 2016) were used for all analyses. We applied a significance level of 0.05 and a two-tailed probability in the correlation analyses.

## Ethical procedure

This study was carried out according to the Declaration of Helsinki as revised in 1989 and was approved by the ethics committee of TMMH on March 7, 2008.

## Results

### Description of the sample

A total of 108 patients (60 male and 48 female, with an average age (SD) of 51.8 (13.9) years) participated in this study. Their demographic and clinical data are shown in Table 1.

The majority of subjects were diagnosed with schizophrenia and were inpatients (92% and 77%, respectively). They had a history of long treatment periods, with an average illness duration (SD, Median) of 23.7 (13.7, 23) years and a median hospitalization frequency of 3. They exhibited moderately severe psychotic symptoms, with an average PANSS total score (SD) of 81.7 (13.3). Significant correlations between clinical characteristics and sex and age are presented in the note for Table 1.

### DOAI assessment

The results of DOAI assessments and the reliability tests are shown in Table 2.

**Table 1. Demographic and clinical characteristics of the subjects.**

| | | |
|---|---|---|
| **Male/Female (%)** | 60 (56) | 48 (44) |
| **Age at investigation (years) (mean (SD), range)** [a] | 51.8 (13.9) | 25–79 |
| **Diagnosis (schizophrenia/schizoaffective disorder) (%)** | 99 (92) | 9 (8) |
| **Inpatient/Outpatient (%)** | 83 (77) | 25 (23) |
| **Age at onset (years) (mean (SD), range)** [a] | 28.1 (10.7) | 13–64 |
| **Education (years) (mean (SD), range)** | 12.2 (2.8) | 6–19 |
| **Marital status (%) (ever married /cohabiting with spouse or partner)** [b] | 26.9 (26.9) | 7 (6.5) |
| **Employment (%) (full-time/part-time)** | 14 (13.0) | 15 (13.9) |
| **Lifetime hospitalizations (mean (SD), range)** [c] | 4.2 (4.1) | 0–22 |
| **Dose of antipsychotic medication (chlorpromazine equivalent dose (mg/day)) (mean (SD), range)** | 587 (457) | 25–2125 |
| **PANSS positive factor score (mean (SD), range)** | 13.6 (2.8) | 7–21 |
| **PANSS negative factor score** | 17.6 (4.8) | 7–31 |
| **PANSS disorganized/concrete factor score** [d] | 8.2 (2.6) | 3–16 |
| **PANSS excited factor score** | 8.5 (2.6) | 4–15 |
| **PANSS depressed factor score** [d] | 7.2 (2.6) | 3–14 |

Note for Table 1: PANSS factor scores are composite scores calculated from the consensus factor model of Wallwork et al. [38].

[a] Female subjects had higher ages at investigation and at onset of the disorder than males (48.0 (13.5) vs. 56.6 (13.1), $F_{1, 106}$ = 11.040, P = 0.001, and 32.1 (12.1) vs. 25.0 (8.2), $F_{1, 106}$ = 13.136, P < 0.001, respectively).

[b] The percentages of ever married subjects and those cohabiting with their spouse or partner for females were higher than those for males (45.7% vs. 11.6%, p < 0.001, Fisher's exact test, and 14.6% vs. 0.0%, p = 0.003, exact test, respectively).

[c] 25th, 50th, and 75th percentiles of the number of hospitalizations in the lifetime history of subjects were 2, 3, and 5, respectively.

[d] Fifty-seven older subjects (older than the average age, 51 years old) had a higher average score on the PANSS disorganized/concrete factor and a lower average score on the PANSS depressed factor than 51 younger subjects (9.00 (2.77) vs. 7.51 (2.21), $F_{1, 106}$ = 9.390, p = 0.003, ANOVA, and 6.21 (2.28) vs. 8.25 (2.61), $F_{1, 106}$ = 17.888, p < 0.001, ANOVA, respectively).

**Table 2. Delusion and its Origin Assessment Interview (DOAI) item scores and the results of their reliability tests.**

| | Mean (SD) | | Clearly present [a] (%) | | Test-retest reliability [b] | Inter-rater reliability [b] |
|---|---|---|---|---|---|---|
| **A. Themes of delusions** | | | | | | |
| A-1. Persecutory delusion | 3.44 | (0.95) | 98 | (90.7) | 0.737 | 0.707 |
| A-2. Grandiose delusion | 1.98 | (1.15) | 37 | (34.3) | 0.956 | 0.951 |
| A-3. Hypochondriacal delusion | 1.71 | (1.07) | 23 | (21.3) | 0.780 | 0.949 |
| A-4. Being assisted or loved delusion [c] | 1.96 | (1.05) | 39 | (36.1) | 0.799 | 0.861 |
| A-5. Delusion of guilt | 1.25 | (0.65) | 5 | (4.6) | 0.728 | 0.792 |
| A-6. Delusion of jealousy | 1.2 | (0.53) | 6 | (5.6) | 0.730 | 0.800 |
| **B. Primary delusions (PDs) [d]** | | | | | | |
| B-1. D-Percep | 2.95 | (1.19) | 79 | (73.1) | 0.885 | 0.661 |
| B-2. D-Memo | 2.25 | (1.27) | 51 | (47.2) | 0.851 | 0.577 |
| B-3. D-Mood | 2.03 | (1.18) | 43 | (39.8) | 0.418 | 0.842 |
| B-4. D-Intui | 2.11 | (1.27) | 47 | (43.5) | 0.625 | 0.734 |
| Frequency of B-1. D-Percep | 1.88 | (1.02) | 59 | (54.6) | 0.600 | 1.000 |
| Frequency of B-2. D-Memo recall | 1.48 | (0.77) | 38 | (35.2) | 0.711 | 1.000 |
| Frequency of B-3. D-Mood | 1.70 | (1.06) | 43 | (39.8) | 0.207 | 1.000 |
| Frequency of B-4. D-Intui | 1.69 | (1.10) | 38 | (35.2) | 0.474 | 1.000 |
| **C. Response and judgments** | | | | | | |
| C-1. Affective responses | | | | | | |
| C-1-a. Unpleasantness | 3.28 | (1.37) | 73 | (67.6) | 0.810 | 1.000 |
| C-1-b. Anxiety and tension | 3.02 | (1.26) | 65 | (60.2) | 0.758 | 1.000 |
| C-1-c. Excitement and anger | 2.77 | (1.07) | 57 | (52.8) | 0.762 | 1.000 |
| C-2. Preoccupation | 3.29 | (1.25) | 74 | (68.5) | 0.537 | 1.000 |
| C-3. Inability to stop delusional thoughts | 2.97 | (1.51) | 64 | (59.3) | 0.354 | 1.000 |
| C-4. Distress | 3.04 | (1.45) | 59 | (54.6) | 0.563 | 1.000 |
| C-5. Pervasiveness | 2.66 | (0.93) | 59 | (54.6) | 0.799 | 0.929 |
| C-6. Delusional system formation | 3.33 | (0.93) | 79 | (73.1) | 0.834 | 0.745 |
| C-7. Judgment about delusion | | | | | | |
| C-7-a. Conviction | 4.27 | (0.99) | 102 | (94.4) | 0.845 | 0.747 |
| C-7-b. Shareability of delusional thoughts | 2.7 | (1.36) | 59 | (54.6) | 0.684 | 0.951 |
| C-7-c. Reality attribution | 4.35 | (1.06) | 102 | (94.4) | 0.663 | 1.000 |
| C-7-d. Denying mental illness involvement | 4 | (1.36) | 92 | (85.2) | 0.593 | 1.000 |

Note for Table 2: D-Percep: Delusional perception: D-Memo: Delusional memories, D-Mood: Delusional mood, D-Intui: Delusional intuition

[a] Number (%) of those who gave an answer equal to or higher than "3. Mild", excluding frequency items for B-1 to B-4, for which an answer equal to or more often than "2. Approximately once a day".

[b] Reliability was assessed by calculating ANOVA-ICC. All items presented here were rated on 5-point scales.

[c] Female subjects had a higher average score for Item A-4 than males (2.25 (1.18) vs. 1.73 (0.88), $F_{1, 106} = 6.814$, p = 0.010, ANOVA).

[d] In MANCOVA, dichotomized variables for B-1 to B-4 were used. A reliability study of the variables was also conducted. ANOVA-ICCs assessing test-retest and inter-rater reliability for dichotomized scores of items B-1 to B-4 were 0.690 and 0.638, 0.818 and 0.647, 0.365 and .633, and 0.650 and 0.667, respectively.

The 4 most frequent delusional themes reported had percentages ranging between 21% and 91%. One hundred subjects (93%) reported that they had experienced at least one PD explained as an origin of delusions with a "mild" or higher than "mild" rating. The other 8 subjects reported hallucinations or hallucination-like experiences as origins of their delusion: auditory hallucination (7 subjects), tactile hallucination (2 subjects), abnormal visceral sensation (1 subject), and "electricity" (1 subject). For each PD, the percentages of subjects who reported their presence with a "mild" or higher than "mild" rating ranged between 43% and

79%. For each of the frequency items, the percentages of subjects who had PDs currently ("approximately once a day" or more often) ranged between 35% and 55%. Eighty-four percent of the subjects reported that they had at least one current PD.

The ANOVA ICCs shown in Table 2 suggested moderate to excellent reliabilities of the DOAI items, excluding 3 items of Frequency of B-3. D-Mood, C-2. Preoccupation, and the dichotomized variable for B-3. D-Mood exhibited slight or fair test-retest reliability, with ANOVA ICCs of 0.207, 0.354, and 0.365, respectively. The average ANOVA ICCs (SD, range) of DOAI items were 0.681 (0.175, 0.207–0.956) and 0.894 (0.131, 0.577–1.000) for test-retest and inter-rater reliability studies, respectively.

The first factor analytic examination of Section C items extracted 4 factors with eigenvalues greater than 1. The second factor analysis using the data set excluding Item C-7-b, which had insufficient factor loading for all factors in the first analysis, produced a simple factor structure and yielded a favorable fit statistic with a non-significant chi-square value of 13.23 (df = 17, p = 0.721). The variances explained by pre-rotational factors were 32.3%, 19.2%, 11.5%, and 9.4% in order of extraction. The factor structure, and means (SDs) and internal consistency (Cronbach's alpha coefficient) of the composite scales are shown in Table 3.

The first factor was composed of anxiety and tension, unpleasantness, and excitement and anger, and was named negative affective response (NAR). The second consisted of distress, inability to stop thinking, and preoccupation, and was designated as activity disturbance (AD). The 3rd factor was composed of C-7 items concerning insufficient illness recognition and was named lack of insight into delusions (LID). The 4th factor comprised delusional system formation and pervasiveness and was named systematization and generalization (S/G). Composite scales constructed by this factor analysis were used for later analyses.

The correlations among the feature scales were weak, ranging from 0.007 to 0.231, except for a moderate correlation of 0.541 between NAR and AD. All 4 composite scales demonstrated acceptable internal consistency (Cronbach's alpha coefficient >0.7). The scale scores were highly correlated with their factor scores, with Pearson's correlation coefficients ranging between 0.869 and 0.966. Therefore, we named these scales similarly to their factors.

Table 3. The factor structure, means (SDs) and internal consistency of composite scales for DOAI Section C: Response and judgment items.

| | Factor 1: NAR | Factor 2: AD | Factor 3: LID | Factor 4: S/G |
|---|---|---|---|---|
| Anxiety and tension (C-1-b) | **0.875** | 0.293 | -0.026 | 0.105 |
| Unpleasantness (C-1-a) | **0.761** | 0.238 | 0.013 | 0.019 |
| Excitement and anger (C-1-c) | **0.677** | 0.400 | -0.014 | 0.077 |
| Distress (C-4) | 0.329 | **0.747** | 0.001 | 0.153 |
| Inability to stop thinking (C-3) | 0.195 | **0.629** | 0.018 | 0.079 |
| Preoccupation (C-2) | 0.220 | **0.528** | 0.062 | 0.045 |
| Conviction (C-7-a) | -0.074 | 0.184 | **0.765** | 0.069 |
| Reality attribution (C-7-c) | 0.082 | -0.009 | **0.708** | 0.139 |
| Denying mental illness involvement (C-7-d) | -0.022 | -0.053 | **0.577** | 0.077 |
| Delusional system formation (C-6) | 0.037 | 0.179 | 0.108 | **0.977** |
| Pervasiveness (C-5) | 0.075 | 0.050 | 0.150 | **0.554** |
| Means (SD) of composite scale scores | 9.06 (3.30) | 9.30 (3.40) | 12.62 (2.73) | 6.00 (1.66) |
| Cronbach alpha coefficient | 0.862 | 0.732 | 0.706 | 0.725 |

Note for Table 3: DOAI: Delusion and its Origin Assessment Interview, Factor loadings greater than 0.5 are indicated in bold.

Cronbach alpha coefficients were calculated to evaluate the internal consistency of the composite factors.

NAR: negative affective responses, AD: activity disturbance, LID: lack of insight into delusions, S/G: systematization / generalization

ANOVA ICCs of the 4 composite scale scores reflected their favorable test-retest and inter-rater reliabilities: 0.837 and 1.000, 0.563 and 1.000, 0.854 and 0.964, and 0.856 and 0.933, respectively.

## Correlation analyses of DOAI and PANSS scores

The results of the correlation analysis of DOAI scale scores with scores of the PANSS consensus factors [38] are presented in Table 4.

As shown in these table, there were many moderate or higher than moderate correlations (with an absolute value of coefficients greater than 0.3 ($p < 0.002$)) among the examined variables. Concerning delusional themes, persecutory delusions had positive correlations with D-Percep, NAR, AD, LID, and with the PANSS excited and depressed factors, and a negative correlation with being assisted or loved delusions. Grandiose delusions had positive correlations with being assisted or loved delusions, D-Memo, D-Intui, LID, and the PANSS positive factor, and a negative correlation with NAR. Being assisted or loved delusions had a positive correlation with the PANSS positive factor and a negative correlation with NAR. These correlations of delusion themes with other variables conformed to their clinical meanings.

Concerning correlations of PDs, excluding those with delusion themes, D-Memo had a positive correlation with LID, D-Mood had a positive correlation with NAR, and D-Intui had positive correlations with the PANSS positive and disorganized/concrete factors. In addition, NAR had positive correlations with AD and the PANSS depressed factor, AD had a positive correlation with the PANSS depressed factor, LID had a negative correlation with PANSS depressed factor, and S/G had positive correlations with the PANSS positive and excited factors.

The correlation coefficients of DOAI variables with individual PANSS item scores were consistent with the connotations of the item names. Thus, persecutory delusions had a strong correlation of 0.685 with Item P6. Suspiciousness/persecution; grandiose delusions and being assisted or loved delusions had correlations of 0.827 and 0.506 with Item P5. Grandiosity, respectively; hypochondriacal delusions had a correlation of 0.682 with G1. Somatic concern; and LID had a moderate correlation of 0.345 with Item G12. Lack of judgment and insight.

Levels of explanation by PDs were significantly correlated with their frequency. The Spearman's correlation coefficients of B-1 to B-4 item scores with their respective frequency scores ranged from moderate to strong: 0.584, 0.689, 0.899, and 0.830, respectively.

The correlation analyses of DOAI variables confirmed their convergent validity and accordingly demonstrated their clinical relevance. Taken together with the favorable results of reliability tests, the DOAI assessment had sound psychometric properties.

## MANCOVA regression analyses

MANCOVA regression analyses of the delusion themes and features with 4 PDs as factors are shown in Tables 5 and 6.

As shown in the table, significant associations identified by the analyses of delusion themes included D-Percep and D-Mood with persecutory delusions, D-Memo with grandiose delusions, and D-Intuit with grandiose, hypochondriacal, and being assisted or loved delusions. In addition, there was a negative association between D-Percep and being assisted or loved, and between D-Mood and hypochondriacal delusion.

The main effects of the 4 PDs on the combined dependent variables of delusional themes were all significant ($F_{4, 98} = 5.150$, $p = 0.002$; Wilks' Lambda = 0.865, $F_{4, 98} = 5.094$, $p = 0.048$; Wilks' Lambda = 0.003, $F_{4, 98} = 5.305$, $p = 0.002$; Wilks' Lambda = 0.862 and $F_{4, 98} = 6.186$, $p = 0.001$; Wilks' Lambda = 0.842, respectively).

**Table 4. Correlation analysis of Delusion and its Origin Assessment Interview (DOAI) scores and PANSS factor composite scores.**

| | A-2 | A-3 | A-4 | B-1 | B-2 | B-3 | B-4 | F1 NAR | F2 AD | F3 LID | F4 S/G | Positive factor | Negative factor | Disorganized / concrete factor | Excited factor | Depressed factor |
|---|---|---|---|---|---|---|---|---|---|---|---|---|---|---|---|---|
| A-1. Persecutory D | -.180 | .124 | -.424** | .413** | .173 | .264** | .029 | .565** | .356** | .019 | .411** | -.108 | -.101 | -.146 | .319** | .335** |
| A-2. Grandiose D | 1.000 | -.012 | .517** | -.048 | .393** | -.172 | .321** | -.421** | -.168 | .357** | .252** | .602** | -.043 | .123 | .177 | -.209* |
| A-3. Hypochondriacal D | | 1.000 | -.076 | -.091 | .033 | -.194* | .182 | .022 | .192* | .089 | .205* | .126 | .189 | .251** | .098 | .064 |
| A-4. Being assisted or loved D | | | 1.000 | -.277** | .140 | -.075 | .234* | -.421** | -.099 | .267** | .046 | .471** | -.101 | .055 | -.156 | -.251** |
| B-1. D-Percep | | | | 1.000 | .119 | .034 | -.070 | .226* | .013 | -.034 | .298** | -.188 | -.251** | -.109 | .273** | .208* |
| B-2. D-Memo | | | | | 1.000 | -.036 | .099 | .032 | .040 | .330** | .216* | .122 | -.209* | -.211* | .147 | .018 |
| B-3. D-Mood | | | | | | 1.000 | .166 | .303** | .147 | -.124 | .053 | -.201* | -.011 | -.133 | -.039 | .272** |
| B-4. D-Intui | | | | | | | 1.000 | -.174 | .037 | .082 | .298** | .302** | .137 | .313** | .198* | .086 |
| F1. Negative affective response (NAR) | | | | | | | | 1.000 | .536** | .003 | .165 | -.236** | -.051 | -.138 | .185 | .377** |
| F2. Activity disturbance (AD) | | | | | | | | | 1.000 | .067 | .232* | .174 | .000 | .014 | .235* | .376** |
| F3. Lack of insight into delusions (LID) | | | | | | | | | | 1.000 | .226* | .269** | .040 | .130 | .053 | -.322** |
| F4. Systematization / generalization (S/G) | | | | | | | | | | | 1.000 | .363** | -.094 | .011 | .351** | .280** |

Note for Table 4: PANSS factors are derived from Wallwork et al. [38]. * p < 0.05, ** p < 0.01

D: Abbreviation of "delusion." F: Abbreviation of "factor."

D-Percep: Delusional perception: D-Memo: Delusional memories, D-Mood: Delusional mood, D-Intui: Delusional intuition

PANSS: Positive and Negative Symptom Scale

**Table 5. Multivariate analyses of covariance (MANCOVA) regression of DOAI delusion themes with primary delusions (PDs).**

| Delusional theme | Origin experience [a] | B | P | 95% Confidence Interval | | Partial Eta |
|---|---|---|---|---|---|---|
| | | | | Lower bound | Upper bound | Squared |
| **Persecutory delusion** | **D-Percep** | 0.715 | 0.000 | 0.330 | 1.101 | 0.118 |
| | **D-Memo** | 0.261 | 0.131 | -0.079 | 0.601 | 0.022 |
| | **D-Mood** | 0.385 | 0.034 | 0.029 | 0.741 | 0.044 |
| | **D-Intui** | 0.007 | 0.967 | -0.338 | 0.352 | 0.000 |
| **Grandiose delusion** | **D-Percep** | -0.133 | 0.573 | -0.599 | 0.333 | 0.003 |
| | **D-Memo** | 0.662 | 0.002 | 0.250 | 1.073 | 0.092 |
| | **D-Mood** | -0.421 | 0.056 | -0.852 | 0.010 | 0.036 |
| | **D-Intui** | 0.657 | 0.002 | 0.240 | 1.074 | 0.088 |
| **Hypochondriacal delusion** | **D-Percep** | 0.143 | 0.533 | -0.598 | 0.311 | 0.004 |
| | **D-Memo** | -0.006 | 0.978 | -0.407 | 0.395 | 0.000 |
| | **D-Mood** | -0.506 | 0.019 | -0.926 | -0.085 | 0.053 |
| | **D-Intui** | 0.528 | 0.011 | 0.122 | 0.935 | 0.062 |
| **Being assisted or loved delusion** | **D-Percep** | -0.472 | 0.031 | -0.900 | -0.043 | 0.045 |
| | **D-Memo** | 0.339 | 0.078 | -0.039 | 0.717 | 0.030 |
| | **D-Mood** | -0.138 | 0.490 | -0.534 | 0.258 | 0.005 |
| | **D-Intui** | 0.448 | 0.022 | 0.065 | 0.831 | 0.051 |

Note for Table 5: Regression coefficients of covariates [sex and age (years)] were not significant.

DOAI: Delusion and its Origin Assessment Interview, D-Percep: Delusional perception: D-Memo: Delusional memories, D-Mood: Delusional mood, D-Intui: Delusional intuition

[a] The reference category was set to "0 (none or unclear)."

**Table 6. Multivariate analyses of covariance (MANCOVA) regression of MOAI delusional features with primary delusions (PDs).**

| Delusion features | Origin experience [a] | B | P | 95% Confidence Interval | | Partial Eta |
|---|---|---|---|---|---|---|
| | | | | Lower bound | Upper bound | Squared |
| **Negative affective response (NAR)** | **D-Percep** | 0.720 | 0.307 | -0.672 | 2.112 | 0.010 |
| | **D-Memo** | 0.518 | 0.404 | -0.709 | 1.746 | 0.007 |
| | **D-Mood** | 2.022 | 0.002 | 0.735 | 3.308 | 0.088 |
| | **D-Intui** | -1.394 | 0.029 | -2.639 | -0.150 | 0.047 |
| **Activity disturbance (AD)** | **D-Percep** | -0.472 | 0.532 | -1.965 | 1.021 | 0.004 |
| | **D-Memo** | 0.663 | 0.320 | -0.654 | 1.980 | 0.010 |
| | **D-Mood** | 1.055 | 0.133 | -0.325 | 2.435 | 0.022 |
| | **D-Intui** | 0.259 | 0.702 | -1.077 | 1.594 | 0.001 |
| **Lack of insight into delusions (LID)** | **D-Percep** | -0.210 | 0.723 | -1.379 | 0.960 | 0.001 |
| | **D-Memo** | 1.593 | 0.003 | 0.562 | 2.625 | 0.085 |
| | **D-Mood** | -0.500 | 0.361 | -1.581 | 0.581 | 0.008 |
| | **D-Intui** | 0.338 | 0.522 | -0.707 | 1.384 | 0.004 |
| **Systematization / Generalization (S/G)** | **D-Percep** | 0.889 | 0.009 | 0.223 | 1.556 | 0.065 |
| | **D-Memo** | 0.246 | 0.408 | -0.342 | 0.834 | 0.007 |
| | **D-Mood** | -0.393 | 0.209 | -1.009 | 0.223 | 0.016 |
| | **D-Intui** | 0.939 | 0.002 | 0.343 | 1.535 | 0.088 |

Note for Table 6: Regression coefficients of covariates [sex and age (years)] were not significant.

DOAI: Delusion and its Origin Assessment Interview, D-Percep: Delusional perception: D-Memo: Delusional memories, D-Mood: Delusional mood, D-Intui: Delusional intuition

As shown in Table 6, significant associations identified by the analysis of delusion features included D-Percep and D-Intuit with S/G, D-Memo with LID, and D-Mood with NAR. Furthermore, there was a negative association between D-Intui and NAR.

Similarly to the findings for delusion themes, the main effects of the 4 PDs on the combined dependent variables of delusion features were all significant ($F_{4, 98} = 2.887$, $p = 0.026$; Wilks' Lambda = 0.895, $F_{4, 98} = 2.487$, $p = 0.048$; Wilks' Lambda = 0.908, $F_{4, 98} = 3.525$, $p = 0.010$; Wilks' Lambda = 0.874 and $F_{4, 98} = 5.053$, $p = 0.001$; Wilks' Lambda = 0.829, respectively).

## Discussion

Based on self-reflections about the origins of delusional experiences enquired about in a semi-structured interview, the present study demonstrated that the majority of 108 schizophrenic/schizoaffective patients identified PDs as origins of the delusions they currently held. This supports the PD-based hypothesis of delusion development, that PDs are delusion-preceding phenomena representing the early stages of delusion development [3, 10, 29]. A unique aspect of this study was that it approached the primary experience of delusions by interrogating such experiences directly, whereas previous studies addressed this issue indirectly, for example examining cognitive features, thought abnormality and affective changes presumed to be related to delusions [2, 3, 7, 13]. The present study also identified individual relationships between PDs and the characteristics of fully developed delusions, further strengthening the concept of that progression to delusion arises secondarily to PDs.

### PDs and delusion progression

The MANCOVA regression analyses in this study revealed multiple associations between PDs and delusion themes and features. They can be interpreted as individual lines of delusion progression processes.

The analysis of delusion themes found two groups of associations with PDs, one between D-Percep and D-Mood and persecutory delusions, and the other between D-Intui and/or D-Memo and grandiose, hypochondriacal, and delusions of being assisted or loved. These two types of associations are easily recognizable in case vignettes described in previous descriptive accounts. For example, 4 of the 6 case vignettes of delusional perception by Jaspers [12] included features of persecutory delusion or ideas, whereas other delusional themes were absent, and both of his case vignettes concerning D-Intui and D-Memo [12] included features of grandiosity.

These types of progression are comprehensible from two points of view: their location (external vs internal) and their thought process characteristics. Persecutory delusions often have external origins, as reflected by their associations with D-Percep and D-Mood. The thought processes involved are typically also related to abnormal interpretation of external stimuli, hence the use of terms like delusional interpretation [2, 6] and paranoid interpretation [39]. In addition, the present study found that persecutory delusions were significantly correlated with negative affects such as anxiety and depression, as also found in previous studies [13, 40]. Therefore, this progression may be considered externally initiated and thereafter internally invading.

In contrast, a second type of progression can be considered to be initiated and reinforced by internal processes. Grandiose, hypochondriacal, and delusions of be being assisted or loved were found to be associated with D-Intui and/or D-Memo, internally evolving types of PDs, but not with D-Percep and D-Mood. Notably, delusions of being assisted or loved and hypochondriacal delusions had negative associations with D-Percep or D-Mood, which suggests that the progression of these delusions requires some degree of disregard of stimuli from the external environment. These delusions were also correlated with high degrees of confidence

(lack of doubt) in the beliefs. This type of progression may also be related to disturbed internal mentation, as suggested by the correlation of D-Intuit with the PANSS disorganized/concrete factor. This factor is related to disturbance in concept integration and concentration [38], which makes it different from that of paranoid interpretation referred to above. In addition, D-Memo, which was found to be associated with grandiose delusions, has been considered to be a phenomenon that is maintained internally by recall repetition even though its initiating experience may have had external origins [3]. Overall, this second type of progression can be considered to be an internally evolved and thereon consolidated process.

The connections of PDs with delusion features found in this study are also of interest. Based on MANCOVA regression, S/G, a feature of highly-developed delusions, was related to D-Percep and D-Intui, suggesting that both external percept-initiated and internally-evolved processes have additive effects on S/G. A closely similar point, that D-Percep, D-Intui, and thought disorder are important in delusion system generation, was stressed by Schneider [1].

The association of D-Mood with NAR may be explained by previous descriptions that D-Mood is characterized by intense negative affect [12, 14]. NAR can therefore also be interpreted as being derived from D-Mood.

In the analysis of subjects' judgments about their delusions, only D-Memo influenced LID, whereas the judgments were alterable by different factors [36]. One explanation for this finding could be that delusional memory is often held more firmly and endurably than other PDs, which may result in its having a stronger influence on the judgements.

These findings of the associations presented above can be interpreted in terms of intensification and generalization of cognitive and affective characteristics recognized in PDs. The important point here is that the formative process of secondary delusions proceeds while maintaining continua with abnormalities inherent to each PD.

The findings are also consistent with those of the cognitive models referred to in the Introduction. The distinction of different cognitive mechanisms underlying persecutory delusions by Freeman and co-workers [39, 40] and Garety and co-workers' model of grandiose delusions [41] has parallels with the delusion theme types found in this study. Similarly, Moritz et al's [31] two-stage cognitive model corresponds in many ways to a progression from PDs to secondarily developed delusions. In this sense, PDs may represent cognitive alterations postulated by cognitive models to be factors for delusion formation.

## Clinical relevance of PDs

The progression processes extracted in this study can be used to re-capture the composition of delusional experiences. The first point to make here is that PDs may reflect the underlying pathology of delusion more distinctively and simply than developed delusions with their complex pathological manifestations. Relatedly, PDs may play a role in the perpetuation of delusions because they are concurrently experienced by patients with their related delusions.

Secondly, this study suggests that there is a wide variation in the progression processes and composing factors of delusions of persecutory and grandiose delusions, pathological affective responses and cognitive disturbance such as delusional systematization and lack of insight into delusion. Therefore, PDs could be of considerable clinical importance for evaluating patients with psychotic disorders and creating effective treatment strategies. In particular, they might be utilized as targets for psychosocial treatment to disentangle the complicated experience of delusion.

Lastly, the findings of this study can be interpreted in the neurobiological context of the aberrant salience hypothesis. Although Kapur [32] and other authors [42, 43] solely highlighted D-Mood as typical of the phenomena, the other three PDs examined in this study can also be considered to involve attribution of "aberrant salience" to particular stimuli such as

external percepts, recalled memories, and internally evolving ideas. Therefore, this study supports van Os [33], who argued that aberrant salience is an essential pathology found in a variety of psychotic experiences. Kapur [44] further maintained that the generally observed delay of response to antipsychotic medication in the treatment of psychoses can be explained by the hypothesis that standard assessment of psychotic symptoms catches only the secondary change after the dissolution of pathological dopamine system regulation. Based on this study, it is possible that PDs might represent a more sensitive indicator to predict the development and decline of delusions than other psychotic symptoms included in standard assessments.

## Limitations

There are several important limitations of the present study. First, the investigation was cross-sectional in nature and therefore was unable to determine causal or temporal relationships among the variables examined. In addition, recall biases may have affected the results. Thus, it needs to be stressed that the term progression used in this study only represented connections in patients' self-reflections elicited by interviews. Second, hallucinations and bizarre delusions [35] were not addressed in the study, although they may act as primary symptoms for subsequent delusional experiences along with PDs [12, 14]. This could be a topic for future studies. Third, caution is needed regarding the sufficiency of PD assessment in this study because it did not thoroughly examine PD-accompanied characteristics, such as un-understandability and abnormality in the form of experience. To make the assessment more comprehensive, these characteristics in addition to those of subjective experiences could be included in future studies. The fourth limitation is that the method of description applied in this study heavily depended on the self-reflections of patients. Jaspers argued that one prominent feature of PDs was un-understandability, which is almost impossible to evaluate without invoking interpersonal considerations. The criticism of Sass and Parnas [45], that cognitive models pursuing understanding from a general psychological viewpoint disregard the un-understandable area of psychotic experiences, is also relevant here. Future studies might address this limitation by incorporating other approaches such as those focusing on the relatedness of PDs to society or other people.

## Conclusions

This study aimed to investigate the key position of PDs as origins of delusions and their close relationship with delusion characteristics, and provided support for this hypothesis, which was proposed by Karl Jaspers over a hundred years ago. This study also revealed many associations between individual PDs and delusion characteristics, which can be plausibly understood as reflecting progression processes from the former to the latter. It is also suggested that further reappraisal of Jaspers' psychopathological theses will provide novel insights valuable for present-day psychiatric practice and research. In addition, this study supports the aberrant salience hypothesis, that phenomena of abnormal salience attributed to some particular stimuli are the underlying pathology of psychoses. Taking these considerations together, PDs may also have an important role in the formation and maintenance of delusions, and could potentially be utilized clinically to sense changes in delusion activity and to improve the effectiveness of psychosocial interventions. Future studies are required to clarify delusion progression processes concerning PDs and to substantiate their clinical meanings.

## Supporting information

**S1 Text. Delusion and its Origin Assessment Interview.**
(DOCX)

**S1 Dataset. Data of DOAI.**
(XLSX)

**S2 Dataset. Data of DOAI reliability study.**
(XLSX)

## Author Contributions

**Conceptualization:** Naoki Hayashi, Yoshito Igarashi.

**Data curation:** Naoki Hayashi.

**Formal analysis:** Naoki Hayashi.

**Funding acquisition:** Naoki Hayashi.

**Investigation:** Naoki Hayashi, Yoshito Igarashi.

**Methodology:** Naoki Hayashi.

**Project administration:** Naoki Hayashi.

**Resources:** Yoshito Igarashi.

**Validation:** Naoki Hayashi.

**Visualization:** Naoki Hayashi.

**Writing – original draft:** Naoki Hayashi.

**Writing – review & editing:** Naoki Hayashi, Yoshito Igarashi, Hirohiko Harima.

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
