## [Decision Letter · Decision Letter 0]

6 Nov 2020

PONE-D-20-21497

Delusion progression process viewed from the primary delusion concept of Karl Jaspers

PLOS ONE

Dear Dr. Hayashi,

Thank you for submitting your manuscript to PLOS ONE. After careful consideration, we feel that it has merit but does not fully meet PLOS ONE’s publication criteria as it currently stands. Therefore, we invite you to submit a revised version of the manuscript that addresses the points raised during the review process.

We look forward to receiving your revised manuscript.

Kind regards,

Raoul Belzeaux, M.D.

Academic Editor

PLOS ONE

Journal Requirements:

2. Please describe in your methods section how capacity to consent was determined for the participants in this study.

3. Please modify the title to ensure that it is meeting PLOS’ guidelines (https://journals.plos.org/plosone/s/submission-guidelines#loc-title). In particular, the title should be "specific, descriptive, concise, and comprehensible to readers outside the field" and in this case  it is not informative and specific about your study's scope and methodology.

Reviewers' comments:

Reviewer's Responses to Questions

**Comments to the Author**

1. Is the manuscript technically sound, and do the data support the conclusions?

Reviewer #1: Partly

Reviewer #2: Partly

2. Has the statistical analysis been performed appropriately and rigorously? 

Reviewer #1: Yes

Reviewer #2: Yes

3. Have the authors made all data underlying the findings in their manuscript fully available?

Reviewer #1: Yes

Reviewer #2: Yes

4. Is the manuscript presented in an intelligible fashion and written in standard English?

Reviewer #1: Yes

Reviewer #2: Yes

5. Review Comments to the Author

Reviewer #1: The authors propose in this study to validate the hypothesis of primary delusion (PD), in the Jaspersian sense, in the development of delusional disorders in patients with schizophrenia. To do so, they propose a self-assessment of delusional forms and contents by the DOAI scale, in a population of 108 patients, without longitudinal evaluation.

The methodology and the writing of the manuscript do not pose any particular problem.

The fine characterization of the links between types of mechanisms (or forms), and types of themes (or contents) is obviously interesting, even without a longitudinal approach, without taking into account hallucinatory processes (as correctly mentioned buy authors) and even without comparison with non-schizophrenic psychotic patients.

However, I have real difficulties with the interpretative scope of the results described, and more precisely with the definition of primary delusions (PD) by Karl Jaspers.

Before proposing a study to “well-verify” the Jaspersian hypothesis, it is advisable, in my opinion, to define it in more detail (introduction and discussion) and more correctly.

If my understanding of Jaspers' specifically schizophrenic delusional processes is correct (especially in ” Causal and Understandable Relationships” section), PD experiences involve several points :

- a break in understanding (incomprehensibility), whether historical, or biographical (through the sequencing of experiences, by “genetic” understanding) or more immediate and empathetic (through “static” understanding),

- This idea of incomprehensibility, which is, moreover, widely criticized by the authors of post-Jaspersian psychopathology, is above all described in an intersubjective mode. In this sense, the idea of self-evaluation by patients, interesting in itself, does not seem to me to be here the most relevant way of evaluating the presence or absence of PD,

- the absence of such PD in patients with non-schizophrenic psychotic disorders (persistent disorders or paranoia, affective disorders), whereas patients with schizophrenia may present both PD and delusion-like ideas (secondary delusions). These different points are briefly mentioned in the introduction, but too quickly, whereas the interpretation and use of the notion of PD appears problematic in my opinion.

Other points, less crucial, seem to me to be important to underline:

- what consideration and what links with PD of the bizarre characteristic of delusion, present in the criteria for schizophrenia in DSM IV (used in the present study), and its absence in DSM 5. A few words on this point could be relevant to the question of Jaspersian incomprehensibility

- the distinction between internal / external delusions, defended in the introduction, has also been widely contested in psychopathology (see Parnas and Sass, 2001, rightly cited here but unfortunately insufficiently integrated)

- the mutualization of patients suffering from schizophrenia and schizoaffective disorder in the sample (I will propose to evaluate only patients with schizophrenia, as long as the presence or not of phenomena labeled as PD is not more robustly evaluated in affective disorders with psychotic characteristics,

- and as recalled by the authors as limits, the absence of a longitudinal approach, the lack of consideration of hallucinatory phenomena.

For all these reasons, and in spite of the interest of a first-person description of delusional phenomena, with the articulation contents / mechanisms, I cannot validate a publication of this manuscript without major revisions, with the following suggestions:

- either a less strictly Jaspersian orientation of the central hypothesis (except for the description of the section B criteria section B of the DOAI scale), to reorient more modestly the whole on the links between the different characteristics of schizophrenic delusions

- and / or a more detailed introduction on the concepts of primary delusion, to allow authors to defend their hypothesis more rigorously.

Reviewer #2: “This phenomenologically inspired article is very well documented and integrates the classical conceptions of Jaspers' phenomenology, here especially the notion of primary delirium, with current neurobiological and psychopathological conceptions such as the notion of Salience found in Kapur or that of ipseite borrowed from Sass. One of the strong points of the article is to highlight how delusions and their themes are progressively formed from an initial perceptual and atmospheric experience (PD) by proposing a correlation analysis comparing data from a semi-structured interview (DOAI) with the most classic schizophrenia assessment scale, the PANSS. It is regrettable that the PANSS was chosen as a basis for comparison because its clinical relevance remains purely symptomatic and descriptive, and has been demonstrated above all in the field of pharmacology, uncritically equating the pathology of schizophrenia with what Jaspers called a cerebral mythology. In the same line, it is regrettable that the ipseity evaluation scale (EASE) proposed by Parnas, or that of Fuchs examinating the world experience (EAWE) closer to a constitutive phenomenology, was not discussed by the authors to complete the fine descriptive and dynamic analysis of the first delusional symptoms. These reservations once expressed, the article nevertheless seems interesting for publication in the context of a re-reading of phenomenological conceptions in the light of recent data from neurobiology and cognitive psychology because it brings to the forefront the crucial question of the formation of delusional symptoms, that of the possible link of meaning between the delusional content and its form.

The article therefore deserves to be published, with minor modifications seeking to further clarify the essential link between the initial delirious atmosphere and its subsequent encystation in the form of persistent delusion. The authors themselves make a close remark about the limits of their study. We suggest the authors might add a few more lines explicating a little more the actuality of the work of Jaspers to emphasize more and more its rich phenomenological posterity as a clinical epistemology bridging subjective experience, cognitive, psychopathological, and ecological conceptions of brain (see Fuchs).

Possibly to be quoted :

Fuchs, T. (2017). Ecology of the brain: The phenomenology and biology of the embodied mind. Oxford University Press.

Sass, L., Pienkos, E., & Fuchs, T. (2017). Other worlds: Introduction to the special issue on the EAWE: Examination of Anomalous World Experience. Psychopathology, 50(1), 5-9. »

6. PLOS authors have the option to publish the peer review history of their article (what does this mean?). If published, this will include your full peer review and any attached files.

Reviewer #1: No

Reviewer #2: **Yes: **Jean Naudin

---

## [Author Response · Author response to Decision Letter 0]

14 Dec 2020

We thank the two reviewers for the very much constructive comments to improve the quality of our manuscript

---

## [Editor Report · Decision Letter 1]

16 Feb 2021

PONE-D-20-21497R1

Delusion progression process from the perspective of patients with psychoses: A descriptive study based on the primary delusion concept of Karl Jaspers

PLOS ONE

Dear Dr/Professor Hayashi,

Thank you for submitting your manuscript to PLOS ONE, and your responses to the reviewers. I have taken over as the academic editor and feel that a number of editorial and other changes still need to be made, as follows:

General

The standard of English is not fully acceptable. If possible, please arrange for the paper to be read and corrected by a native English speaker.

It would make sense to use the term understandable/ununderstandability instead of incomprehensible/incomprehensibility, as this is the term customarily used to refer to Jaspers' thinking on this point.

Introduction

‘However, delusions the patients present with are endo-products of concurrent disturbances of mental activities such as thinking, perception, memory, and affect.[2, 4, 60 6, 7, 9, 10]’

-    This sentence assumes that there are established facts about the origin of delusions. This is not the case.

-     By ‘endo-products’ do you mean ‘end-products’.

‘Bizarre delusions defined in the criteria of the Diagnostic and Statistical Manual of Mental Disorders, 4th Edition (DSM-IV) [16] (though absent in those of DSM-5) and the core symptoms of influence, passivity, or control experiences in the description of the 11th revision of the International Classification of Diseases (ICD-11) were included in this list.[17]’

-    As far as I know bizarre delusions were not included as first rank symptoms by Schneider. Please clarify.

‘The PDs referred to by Jaspers [9] were delusional awareness or mood (atmosphere), delusional perception, and delusional idea (delusional memory and delusional intuition (sudden notion)). [9]’

-    Reference 9 refers to a 1913 version of Jaspers’ book, General Psychopathology. As far as I know Jaspers does not argue that delusional memories are primary delusions in his later, English-translated version of the book. Please justify what you say here, eg by reference the book edited by Fuchs et al that you cite later.

‘Beck et al. [32] proposed a hypothesis that delusions are formed from the impairment of reality testing and cognitive biases such as self-referential and externalizing biases. Similarly, Garety et al. [10] proposed a cognitive model of delusion formation, in which abnormal cognition, emotion, and thought disorder promote its formation. Moritz et al. [33] also presented a 2-stage cognitive model composed of liberal acceptance (belief formation) and consolidation by factors such as emotional state (belief perpetuation).’

-    These sentences refer to work that is essentially outside the clinical descriptive/phenomenological approach to delusions that is the focus of the paper. I would recommend omitting them. Alternatively, if desired you could add a paragraph in the discussion on the degree to which the phenomenological approach and your findings concur with these cognitive/cognitive therapy-based approaches.

'From there emerged the hypothesis that primordial conditions of delusion are connected with a certain neurobiological abnormality. Kapur [34] proposed the aberrant salience hypothesis that chaotic disinhibition of subcortical and prefrontal dopamine release in acute schizophrenic psychosis is associated with excessive salience attributed to otherwise irrelevant stimuli, which is typically noted in the delusional mood. Subsequently, van Os [35] widened this theory and maintained that aberrant salience attribution underlies a wide variety of psychotic symptoms.’

-    The same applies to these sentences. If desired you could say something about Kapur’s theory in the discussion with respect to your findings (which you already do to some extent). Your finding of a very high frequency of referential delusions is clearly relevant here.

Method

‘Scores of PANSS cluster scales: Anergia, Thought disturbance, Activation, Paranoid belligerence, and Depression, were used in analyses.’

-    Most clinicians and researchers who use the PANSS will be familiar with its positive, negative and general psychopathology subscales, and also that factor analysis results in five factors (positive, negative, disorganization, mood elevation and depression, for a review see Wallwork RS, et al. Searching for a consensus five-factor model of the Positive and Negative Syndrome Scale for schizophrenia. Schizophrenia Research. 2012;137:246-50.). However, the ‘PANSS cluster scores’ that you have used are unfamiliar and it is not clear what they are based on. Please elaborate more on the background to this subdivision. You might also consider replacing the parts of the analysis relevant to the PANSS with the more intuitive five factor scores.

Results

‘One hundred subjects (93%) reported that they had experienced at least one PD explained as an origin of delusions with a “Mild” or higher than “Mild” level.’

-    The average duration of illness in the sample was >20  years. It is therefore not entirely clear what you mean when you talk about origins of PDs. Specifically, did the PDs occur recently or long ago?

-    Delusional memory was present at some point in nearly half the sample. This seems very frequent for a symptom that is usually considered to be rare. Please comment on this finding in the discussion.

Discussion

Please review for clarity. In particular, when you make assertions based on findings of the study, it would be helpful to make clear what the relevant findings were. 

Please submit your revised manuscript within three months, if possible. If you will need more time than this to complete your revisions, please reply to this message or contact the journal office at plosone@plos.org. We look forward to receiving your revised manuscript.

Kind regards,

Peter McKenna

Academic Editor

PLOS ONE

---

## [Author Response · Author response to Decision Letter 1]

9 Mar 2021

I appreciate much the editor's comments useful to improve the quality and readability of manuscript.

---

## [Editor Report · Decision Letter 2]

17 Mar 2021

PONE-D-20-21497R2

Delusion progression process from the perspective of patients with psychoses: A descriptive study based on the primary delusion concept of Karl Jaspers

PLOS ONE

Dear Dr./Professor Hayashi,

Thank you for your further revised version. While this deals with most of the points I raised, please could you address a few minor outstanding points in order to enhance the article's readability:

Introduction

‘In addition, Kendler et al. [21] and Oulis et al. [26] performed factor analytic studies on delusion characteristics, and extracted factors, such as delusional involvement and cognitive disintegration; however, their results were not consistent.’

For clarity, please briefly state what these authors meant by  delusional involvement and cognitive disintegration.

‘The PDs referred to by Jaspers were delusional awareness or mood (atmosphere), delusional perception, and delusional idea (delusional intuition (sudden notion) and delusion confirming memory (delusional memory)) [6, 12].’

Please insert a sentence or two based on your response to the editorial comment about delusional memory. For example, this could be something along the lines of: ‘While Jaspers did not use the term delusional memory, but instead referred only to delusional content confirmed by a vivid memory, Sims (reference: Symptoms in the Mind:, 3rd Ed.,2003) later explicitly included delusional memory along with the other three types of PD proposed by Jaspers.’

Method

‘(iii) not having a prominent organic cognitive disorder or intellectual disability that makes the assessment difficult’

Please clarify what you mean by ‘not prominent organic cognitive disorder’

Although the standard of English is now much improved, there are still several instances of unclear meaning or unconventional phrasing. When you have made the above changes, I suggest you send the ms to me as a Word document at the above cc address (mckennapeter1@gmail.com) and I will correct these for you.

Sorry for the continuing delays in acceptance.

Kind regards,

Peter John McKenna

Academic Editor

PLOS ONE
---

## [Author Response · Author response to Decision Letter 2]

11 Apr 2021

We are greatly grateful to Dr, Peter John McKenna, the academic editor for presenting quite constructive comments and providing a myriad of corrections for our manuscript.

We are really astonished by the kindness he has offered to us.

Please find our responses to the three comments in the decision letter on 17th March, 2021 and eight ones revealed in personal e-mail communications in our response report.

---

## [Editor Report · Decision Letter 3]

14 Apr 2021

Delusion progression process from the perspective of patients with psychoses: A descriptive study based on the primary delusion concept of Karl Jaspers

PONE-D-20-21497R3

Dear Dr/Prof Hayashi,

We’re pleased to inform you that your manuscript has been judged scientifically suitable for publication and will be formally accepted for publication once it meets all outstanding technical requirements.

Kind regards,

Peter John McKenna

Academic Editor

PLOS ONE
---

## [Editor Report · Acceptance letter]

16 Apr 2021

PONE-D-20-21497R3 

Delusion progression process from the perspective of patients with psychoses: A descriptive study based on the primary delusion concept of Karl Jaspers 

Dear Dr. Hayashi:

I'm pleased to inform you that your manuscript has been deemed suitable for publication in PLOS ONE. Congratulations! Your manuscript is now with our production department. 

Kind regards, 

on behalf of

Dr. Peter John McKenna 

Academic Editor

PLOS ONE